# Effects of a Personalised Physical Exercise Program on University Workers Overall Well-Being: “UAL-Activa” Program

**DOI:** 10.3390/ijerph191811331

**Published:** 2022-09-09

**Authors:** Antonio Jesús Casimiro-Andújar, Ricardo Martín-Moya, María Maravé-Vivas, Pedro Jesús Ruiz-Montero

**Affiliations:** 1SPORT Research Group (CTS-1024), CERNEP Research Center, Department of Education, Faculty of Education Sciences, University of Almería, 04120 Almería, Spain; 2Physical Education and Sport Department, Faculty of Education and Sport Sciences, Campus of Melilla, University of Granada, 52005 Melilla, Spain; 3Department of Pedagogy and Didactics of Social Sciences, Language and Literature, Universitat Jaume I, 12006 Castellón, Spain

**Keywords:** personal training, physical health, emotions, well-being, physical exercise

## Abstract

*B**ackground and objectives:* Regular individualised physical exercise (PE) is a habit that not only has consequences for overall health (physical, emotional, social and mental) but can also have positive effects on organisations and institutions, as it helps workers to improve their personal balance and recover from the effort of their working day, showing higher levels of energy, commitment and productivity. The aim of this study was to understand the relationship between the practice of PE and well-being in personal life and at work, as well as job satisfaction, overall health and the assessment of the personal training service provided by final-year students studying for a degree in Physical Activity and Sport Sciences. *Methods*: This study used a qualitative research methodology. There were 25 employees of the University of Almeria (UAL) (M = 52.16 ± 9.55 years), divided into two focus groups and participating in the physical exercise program “UAL Activa”. *Results*: The following three main themes, based on the results, emerged: (a) social well-being during PE practice, (b) assessment of the personal training service and (c) physical exercise as an emotional benefit at work. *Conclusion*. The workers’ interventions have shown that participating in a personalised PE program led by a personal trainer can improve overall health and mood, with a very positive influence on the working environment.

## 1. Introduction

The World Health Organization (WHO) [1] estimates that at least 23% of the adult population and 81% of adolescents worldwide have physical activity levels below those needed to maintain health and control their body weight. This lack of activity may be associated with non-communicable diseases, depression and anxiety problems [2], as well as a lack of strength and energy during the working or academic day [3].

A possible solution to such a sedentary and inactive lifestyle, and its negative consequences, would be the targeted practice of Physical Exercise (PE). Acevedo (2012) [4] conceptualises it as “a structured form of physical activity, with the specific aim of improving or maintaining health or fitness” (p. 4). Furthermore, he demonstrates that recurrent and sustained PE can generate physical benefits at the cardio-respiratory, muscular and skeletal levels, as well as reduce the risk of non-communicable diseases [5,6]. Other studies have recurrently shown that PE can also be valuable on a physiological level: PE practice increases levels of endocannabinoids [7,8], endorphins (Hicks et al., 2019), serotonin [9] and dopamine [10]. These neurotransmitters are responsible for pain reduction, emotion regulation, pleasure [11] and stress reduction [12]. Equally, the practice of PE would also be beneficial in improving psychological well-being, thereby helping to provide a mental distraction from the demands of the working day [13], increasing feelings of mastery, increased self-efficacy and recovery from stress levels [14].

Work can be a source of self-fulfilment, but also a scenario of stress, distress and anxiety. Competitiveness, work stress, time pressure and tensions at work favour burnout syndrome, a state of physical and emotional exhaustion caused by overwork, lack of free time, and always working in a situation of urgency [15]. Furthermore, these authors point out that the most important parameter for a company is the human factor, as there is scientific evidence that increasing physical activity has a positive effect on employees, thereby improving their sense of identity and helping the organisation with its vision and mission. This should include promoting physical activity and health in the workplace, improving quality of life, building healthy relationships among all colleagues, reducing absenteeism and improving the work environment.

The concept of well-being is broad and includes various dimensions [16]. Thus, there is eudaimonic and hedonic well-being. Eudaimonic or psychological well-being lies in being able to efficiently perform activities in line with one’s own life values [17]. Hedonic well-being, also called subjective well-being, has the vital objective of accumulating experiences of pleasure [17]. For this reason, effective well-being programs for workers should be holistic and include several components, such as counselling, information, mindfulness, initial assessment and personalised Physical Exercise [18]. Including PE on a regular basis as part of a wellness program is reasonable, appropriate and feasible, as it provides benefits for somatic and psychological well-being [3]; therefore, PE is seen as a leading facilitator in achieving increased well-being [6].

However, a poor state of physical and mental health of workers in any sector creates significant problems in job performance. One of the issues associated with such poor physical and mental fitness is stress [15]. This work-related mental disorder is one of the main causes of absenteeism in Spain, costing more than 10 billion euros for days lost at work, which has a negative impact on productivity [19]. Such figures are equally high worldwide, especially in more “developed” countries such as Australia, the United States and all of Europe [20].

As noted above, the practice of PE also produces benefits on an emotional and psychological level. In their study, Feuerhahn [21] show that on days when workers perform PE during their workday, they perceive an improvement in their physical and mental well-being, understood as the presence of healthier emotional states, thanks to the positive metabolic adaptations they receive after the practice. Moreover, these benefits at the emotional level are projected at the organisational level because there is ample evidence that workers’ positive emotional states are important antecedents of good work outcomes and success [22,23]. Indeed, non-sedentary individuals are more empathetic and less distracted in their work tasks in a university environment [24].

Although there is evidence of the relevance of PE for individuals and, by extension, for organisations, there is little research on the impact of this practice on employee well-being. Therefore, the aim of this study is to understand the relationship between the practice of PE with health and well-being at work, understood as one of the relevant aspects of job satisfaction, the influence of PE on emotions and the assessment of the personal training service when performing PE.

## 2. Materials and Methods

### 2.1. Procedure

This study used a qualitative research methodology. The aim was to collect information from a purposive sample of University of Almeria (UAL) employees participating in the “UAL activa” program regarding their experiences of initiating and maintaining a personalised and supervised physical exercise program. This methodology was appropriate for investigating subjective and personal accounts that require explanation; such experience is well suited to qualitative research [25,26,27]. A qualitative enquiry was well suited as this research aimed at delving deeper into the experiences of participants undertaking personal training by inquiring into their personal perspectives and accounts [27]. Qualitative research is used when open and reflective information is required from research participants to achieve greater depth in the study [28].

### 2.2. Participants

The study was open to all people enrolled in the “UAL Activa” program for workers at the University of Almeria (UAL) who had no medical contraindications to doing PE during the period of participation. A total of 68 members of the teaching and research staff and the administrative staff of the University of Almeria participated in the six-month program. However, only 14 women and 11 men (M = 52.16, SD = ±9.55 years) gave their consent to participate in the present study (focus group).

A total of 12 final year undergraduate students of Physical Activity and Sport Sciences (CAFD) from the University of Almeria who wished to participate, always supervised by a lecturer specialising in physical activity and health from that university, were responsible for designing and carrying out the PE tasks developed with the UAL professionals.

### 2.3. UAL-Activa Program

This is a theoretical–practical learning program for CAFD students, as well as a service to the entire UAL. The main objective was to promote the transfer of knowledge to the university community and to develop learning spaces where the participating university students would take an active part in their learning, as well as receive a training and follow-up program throughout the process.

As a sports activity, all UAL employees were offered, free of charge, a supervised PE program aimed at improving body composition, health-oriented physical condition, quality of life and improvement of the level of physical activity of all those enrolled in an attempt to achieve a more active and healthy university.

This program has been running since 2017 to date, with great acceptance by the university community. The activity is carried out in the fitness room of the university. The university students involved work on a personalised basis with each enrolled worker, who undergoes an initial assessment that details the interests, availability and objectives that each participant seeks through their participation in the program.

The tasks were carried out according to the objectives and level of the user, once a week for 6 months, and each session lasted 60 min. The PE that was worked on in each session were mainly strength by means of auto-loads, elastic bands, weight machines, TRX and free weights, although there were also tasks to improve cardio-respiratory capacity, flexibility, breathing and relaxation, all with controlled and progressive intensity by the students who directed and supervised each session.

### 2.4. Compilation of the Participants’ Observations

Two focus groups were developed with the participation of the main researcher and an assistant for the transcription and annotation of the most important details in real-time. Following Hamui-Sutton and Varela-Ruiz [29], it was considered necessary to use a thematic guide as an instrument for the moderator to follow up on the topics of interest during the development of the focus group, which should be composed of open questions related to the topic to be studied. The role of the moderator was to show an attitude of attentive listening that would help to maintain the workers’ conversation and guide the observations towards participation and relevance in all the issues raised. In this sense, it was important to take into account, in addition to the topics provided in the topic guide, new concepts, ideas or information that could be generated by the group. Some of the questions in the thematic guide were: (a) How do you think your overall health has improved since participating in UAL Activa? (b) How do you think the program has affected your state of mind? (c) How do you think the intervention of the different students -personal trainers- has influenced your professional work in the work environment? and (d) After participating in UAL Activa, how do you evaluate this new personal training program at the University?

Prior to the start of each focus group, before signing an informed consent form, the workers were informed that the conversation would be recorded both on video, to take into account the expressions and body interactions at the time of each intervention, and on audio for subsequent transcription and analysis. The room in which the focus groups were held was designed to facilitate the smooth running of the group, being a comfortable space where the participants were organised in a U-shape for better dynamic communication and provided with water for proper hydration.

### 2.5. Intervention Analysis

It followed the approach of Vasilachis de Gialdino [30], nourished by the contributions of Richardson and Pierre (2017), to interpret the statements and their intersections with the formal and implicit theories of the researcher, being the analysis developed from a holistic point of view. It used the metaphor of crystallization, which refers to the multiple faces that intersected to reflect the different perspectives of the study, having to be aware that these would not be static, but in constant change and movement. Richardson and Pierre [31] (p. 135) tell us that “we do not triangulate, but crystallize, as the central imagery is the crystal, combining symmetry and substance with an infinite variety of shapes, transmutations, multidimensionalities and angles of focus”. In short, what is going to be transmitted is what we see and interpret, but we cannot ignore that it must be a faithful reflection of what is observed from the angle that, as researchers, we occupy in the study.

A mixed methodological option for the analysis was used, starting from the previous questions marked in the focus group script (deductive) and an emerging inductive option, respecting the statements of the participants. We have used some of the guidelines set out in Grounded Theory [32], outlining the statements of the study from the particular to the general, going through the process of categorisation and subsequent coding following a non-linear bidirectional process that enriches the subsequent analysis and categorisation. The general emerges from the data itself, in this case, from the statement of the study participants. The information produced from the video and audio recordings of the focus groups and the notes collected from the observation of each focus group was organised and prepared for processing with Nvivo software (Version 12, QSR International Pty Ltd., Melbourne, Australia). Direct transcription of the audio-visuals with the software allowed us to identify the real timeline of each intervention for further analysis. An open axial coding has been carried out, based on the previous and emerging categories, mainly in the second and third levels of categorisation, in line with the researchers’ interpretation.

The analysis of the data, supported by NVivo software, was mainly based on the coding matrices in order to bring out the relationships between the different categories and cross-reference those that are of interest throughout the analysis and the attributes that make up the criteria of homogeneity and heterogeneity. This made it possible to compare, establish relationships and make visible the questions generated by the participants’ statements. The strategies used enabled us to confront the implicit theories of the participants with those of the researchers, generating the substantive theories of the research.

### 2.6. Ethical Considerations

The ethical criteria and good practices established by the UAL for research have been faithfully followed. In the development of the focus groups, informed consent and permission were sought and freely given by the participants and the management of the Sports Service of the UAL. Following Quinn Patton and Cochran [26], the ethical issues that must be followed in all research, such as confidentiality and consent, were fully complied with. Therefore, participants were duly informed about the objectives and topics to be addressed, preserving their identity above all else. In this sense, each participant is referred to by a pseudonym in order to maintain their anonymity. The study was conducted in accordance with the ethical principles of the Helsinki Declaration for human research and was approved by the Research Ethics Committee of the UAL (code: 2021/66).

## 3. Results

This section mainly presents how the workers perceived and experienced their participation in the “UAL Activa” program. It seeks to understand the relationship between the practice of PE with overall health and well-being at work, valuing this new personal training service in a university context.

### 3.1. Social Well-Being during the Practice of PE

It is well known that university workers spend much of their working time in isolation from their co-workers, so opportunities for social interaction with each other are limited [33]. Participants highlighted that one of the aspects that significantly improved their mental and emotional well-being was social interaction at work and communication between peers, which had been limited to date, and there had always been a desire to increase it. Therefore, this scenario and the “UAL Activa” program provided an ideal environment that was conducive to such interaction. In this context, the intervention of Deva (12) is shown: “The fact of meeting faces that we usually know, but in other contexts, and seeing them there in the gym, well, it was a great joy”. This feeling was also described by Alio (14): “I really like socializing and in our day-to-day life it is more complicated, this is a different space but very satisfying”. Finally, the contribution of Laro (15) was described along these lines: “I definitely gained friendships and I can walk around the University facilities, greet people and call them by name. You connect at a deeper level”.

The UAL Activa program helped workers address the problem of social isolation among colleagues. Participants recognised that the program gave them the opportunity to connect across subject areas, departments and other professionals in the university. Participants established links with their peers in the pursuit of an active lifestyle.

### 3.2. Evaluation of the Personal Training Service in the Work Environment

Regarding the joint reflection on this new health project at the university, a common theme emerged among the participants’ responses. The employees highlighted the fact that people with poor physical conditions get the extra motivation to make a change in their lives and implement a healthier lifestyle, valuing the help of a personal trainer positively. The discourses highlight the fact that people often have the internal motivation but lack the necessary knowledge, which will lead to failure with regard to their goals. In this context, some interventions are detailed: “in my case what I value is personalized attention, because they explain the movement, they adapt the exercises to the person’s needs and this personalization is essential” (Laro, 78).

In this sense, Atta (84) also values the importance of the personal trainer and the ability to learn about what is being done: “What I value most is everything I learnt. I learned a lot from the coach, always with new things and he always told me let’s try this and that”. Likewise, the confidence offered by a service offered by qualified people: “I would also add confidence, the certainty that what you are doing is right and is based on science and health” (Arnua, 82) and “the personalized attention, also the professionalism and confidence” (Neco, 90).

One of the most relevant aspects of supervised training is the participants’ confidence when facing new challenges, correcting bad habits, improving their technique or overcoming previous injuries. Relying on the judgment of a professional seems to have been key to the participants’ mental and physical improvement. These facts can be better understood based on the comments of Atio (91): “How important it is to know how to do the exercises so as not to do anything stupid, not to hurt yourself and to do them well”; Laro (93): “I already had acquired vices from having done sport or exercises incorrectly and he (the trainer) was very interested in correcting me and, therefore, I highly value the professionalism he has shown towards me”. The importance of the trainer’s treatment of injuries and personal health was also highlighted in Deva’s interventions (86): “He adapts to your needs, that is, to small injuries that you have had. In my case, my ankle was quite painful and I had to do exercises to strengthen it” and Epana (87) “I, for example, have loved the challenges that the coach has given me continuously and they have helped me a lot, because I was a bit scared because of an injury and he has always pushed me to the limit, well not to the limit, but he has taken me beyond where I was confident that I could do it, and I have been able to do it. So, it’s really good”.

As can be seen in the comments of the participants, the involvement of personal trainers is valued very positively, as they are key players in informing, correcting and giving the necessary confidence to each participant with the aim of maintaining the practice of sport and, with this, improving the quality of life and well-being of the subject.

### 3.3. Physical Exercise as a Benefit for Emotional Health in the Workplace

In order to investigate how workers improved their well-being and developed resilience against negative aspects such as anxiety and excessive stress, their particular descriptions have been analysed within the focus groups. The participants’ discourses showed that they felt less stressed, more grateful, more in control of their emotions, less physically and emotionally exhausted and better able to manage stressful moments such as the end of the academic year. In this context, the contribution of Arena (103) emerges: “the stress of the end of the academic year is absolutely crazy, but I can say that I have handled it much better this year because I woke up with more energy thanks to physical exercise”. Along the same lines, Laro (99) emphasises the importance of how PE has a positive influence on her emotional well-being: “Well, as I said, I’m nervous and overwhelmed by work, but in general, since I’ve been doing exercise, I’m happier. So good, I feel happier and calmer”.

In addition to reducing stress, it was also a way to improve well-being and a sense of efficacy. For example, Epana’s intervention (107): “It has given me mental strength. I thought I couldn’t be able to do some things I was asked to do and I have been able to do everything perfectly” and the contribution of Atio (112): “In my case better than before. I have felt much more positive in my life. Since I’ve been doing sport, I usually feel much more positive, cheerful and, above all, practical at work”.

Finally, Verna’s (105) intervention shows the relaxing and disconnecting effect of PE and its positive influence on mental and emotional health: “It has made me burn off energy and relax in this busy time of year. Emotionally it’s great for me”. 

## 4. Discussion

The present study seeks to understand the relationship between the practice of PE, holistic health and well-being at work through a personalised PE and wellness coaching program to adopt healthy lifestyle habits and physical practice as part of a lifestyle change. The comments of the participants of this study have demonstrated the effectiveness of this intervention to improve their physical condition, motivate and modify the healthy conduct of the administrative staff and teachers and researchers of the UAL, providing a better perception of stress control and other healthy behaviours that have led to considerable improvements in health habits, awareness creation and improvement of the social aspect among peers. Along these lines, and following Du et al. [34], once people experience the benefits of exercise, they are likely to be inspired to continue the practice and gain benefits.

The exercise intervention in this study took place in the fitness room outside working hours. Therefore, one of the keys to the success of this PE intervention may be due to its practicality, greater convenience, flexibility, and free of charge and autonomous management for UAL employees during their non-working hours. In this context, individual exercise can facilitate better compliance with the proposed objectives [15], as was evident in this study, where adherence and program completion rates were very high.

Emotional well-being, so necessary in these times of social upheaval and mental health disorders, has been shown to be one of the main benefits found by the participants in this program, helping them in their daily lives both personally and professionally. These results could be explained by what Catalino et al. [35] call “prioritization of the positive”. This concept is based on the happiness formula of Lyubomirsky et al. (2005), which states that happiness is composed of genetics, circumstances and intentional activities. Based on this last element, Catalino et al. [35] established that people who organise their daily lives by seeking positivity have more constructive emotions and higher levels of life satisfaction. In the present study, PE has meant for participants with high levels of energy and vitality, willing to invest effort in the activity being developed, even when difficulties appear along the way, such as at the end of the academic year. Therefore, there is both intention and motivation to carry out a healthy activity, promoted by the active role of the participating students as a fundamental aspect of their teaching–learning process.

One possible explanation for the impact of PE on overall well-being is that, through a motivational process, the basic psychological needs of autonomy, relatedness and competence are satisfied, as postulated by Self-Determination theory [36]. Indeed, regular physical exercise planned in a personalised way improves motor competence, while knowing the “why” of each task in terms of its objectives satisfies the need for autonomy while sharing moments of physical exercise with other co-workers satisfies the need for belonging. Thus, people who feel more energised have a sense of energetic and affective connectedness at work and with their peers.

In terms of perceived stress, the analysis of the qualitative findings suggested that participation in the PE program at “UAL Activa” reduced the levels of work-related stress of the administrative staff and teachers and research staff. These findings are in line with other intervention studies in professionals, which have shown statistically remarkable results of stress reduction with various approaches such as meditation, physical exercise, mindfulness or a combination of these [33,37,38].

Indeed, participants in the two focus groups that have been part of the “UAL Activa” program have experienced stress relief and improvements in well-being, expressing gratitude for the service received, feelings of joy or maintaining an optimistic outlook despite challenges; this finding being consistent with previous research on the impacts of physical practice on well-being [39,40]. The nature of the qualitative data could serve to strengthen relationships between PE practice and well-being, as claimed by quantitative studies [41,42].

Finally, participants’ interventions have also shown that physical practice could improve quality of life and holistic health, as previously reported by Dauwan et al. [43]. Such findings on improved well-being are more important for those workers who report stressful episodes and previous low levels of physical activity (inactive and sedentary). The physical practice could counteract the negative effects of common work-related factors associated with psychological health, such as stress and lack of control over work [15]. Interventions of workers have shown that participating in this PE program under the supervision of a personal trainer could improve mood, emotional states and the pro-social aspect of the work environment, as confirmed by other authors [44].

This study has several limitations. As it presents a qualitative approach, the results cannot be generalised. In addition, there are likely to be additional differences among participants that were not captured in the observations but contributed to the results. There is also a limitation around self-selection; administrative staff and teaching and research workers who agreed to participate in the focus groups may not represent the experiences of workers with similar levels of attendance and who did not participate in the present study. The nature of the focus group may influence participants’ responses which could have been quite different if the composition of the group had varied.

## 5. Conclusions

The findings of this study supported the subjectivity of well-being, as each of the participants noted their own individual goals and sources of satisfaction with life and work, including in relation to exercise. The perceived well-being described by the participants in this study had a positive impact on well-being, although each participant experienced it differently. The research findings supported that workers described the benefits of supervised PE from a positive and holistic perspective.

The workplace can become a place fraught with physical and emotional stresses. In the face of this, appropriate physical exercise can reduce such work-related stress and some psychological dysfunctions linked to pressure at work, such as anxiety, depression and other psychosocial illnesses.

Detailed studies are needed to assess the quantification of the benefits for cognition and overall health. However, considering the benefits of exercise in other areas, it seems likely that regular, moderate exercise could improve many aspects of our lives, including performance at work. Further studies are needed to generate long-term positive changes and sustained improvements after PE intervention.

## Data Availability

The datasets generated during and analyzed during the current study are available from R.M.-M. on reasonable request. No individual or indemnifiable data are being published as part of this manuscript.

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
