# Peer review of "Effects of a Personalised Physical Exercise Program on University Workers Overall Well-Being: “UAL-Activa” Program"

_ijerph, 2022, doi:10.3390/ijerph191811331_

Round 1

Reviewer 1 Report

Dear Authors,

You raised a very important social topic of the role of physical activity in the context of the well-being of an individual - an employee and an institution / organization - workplace.

Qualitative research helps to better understand the situation of employees, the work environment and have shown that participating in a personalized PE program led by a personal trainer can improve overall health and mood.

Although the topic is known, it requires constant updating and reminding societies about the role of physical activity in human life.

In my opinion, the work can be published as it stands.

Yours faithfully.

Author Response

Thank you very much for your comments about the study, it is certainly great news that you appreciate it. A cordial greeting.

Reviewer 2 Report

Interesting and clear study. However, we consider that the presentation of more data, its systematization and evidence will be a strong contribution to the enrichment of the study. The presentation of results of a qualitative nature does not prevent more rigor and systematic character in the presentation of results (even collected in focus groups).

Author Response

Parts of the work have been modified to further enrich its content. We hope that, after this new version, we have improved the quality of research. The general feedback has not allowed us to know in which paragraphs exactly we had to modify the text. We hope that with the requests of the rest of the reviewers, the work has improved in quality. You can find in yellow the changes made.

Reviewer 3 Report

Introduction

lines 50-60: This paragraph contains a few run on sentences that are not very clear, and the ideas do not always flow in a fully connected manner within each long sentence. Please review and revise for better clarity.

line 67: Why did you not use the abbreviation PE at the end of this sentence?

lines 85-86: Did you mean to say non-sedentary individuals are more distracted?

line 90: As written, you are operationally defining health and well-being at work as job satisfaction--is that what you meant? It seems like job satisfaction would accompany many things beyond health and well-being

Lines 89-94: it seems like there are multiple purposes here; for clarity, it may be better to break this up into a primary purpose statement and a secondary purpose statement. Also, I think you could leave out the last clause about sample characteristics until methods (though you don't have to if you really want it there).

Methods

lines 154-155: When you write, "the intervention of the different students," are you focusing on the subjects doing the exercise, which just happened to be delivered by students, or are you focusing on subjects interacting with students?

line 173: In your quote, check that the word used is "imaginary" instead of "imagery"

lines 200-207: while it is explicitly stated that informed consent was obtained, the body of the manuscript never mentions that the research was approved by an Institutional Review Board; that is only included in the declaration statements at the end. Typically, this is included in the body text as well.

qualitative methodology is well described and conforms to accepted practices

Results

no suggestions for results section

Discussion

lines 335: by saying "experienced greater stress relief," you imply that you are comparing the experiences of the participants against someone else, but there was no comparison group in this study, nor do the other citations here provide a comparison. I think you may just need to delete the word greater

Conclusion

line 354-355: When you write, "the perceived well-being described by the participants in this study was positive," I don't think that is necessarily accurate. Partly, I don't think your study has the ability to determine if the subjects' well-beings were overall positive. I think maybe you meant the intervention had a positive impact on well-being.

lines 361-362: This sentence seems to express the results that would come from a quantitative study that measured stress and anxiety levels; I don't know that this statement is appropriate to the present research methods

lines 363-364: While there were some individual quotes in which participants stated they felt overworked or stressed out from work, unless an inclusion criteria was subjects had to feel overworked, this may be overextending your results

lines 363-368: this paragraph is better suited in the discussion section than the conclusion section

lines 369-376: limitations belong on the discussion section, not the conclusion section

Author Response

We would like to sincerely thank your review for their thoughtful and constructive comments, which have undoubtedly improved the quality of our manuscript.  We have carefully considered all of the suggestions and have integrated them into the revised manuscript. Changes to the original manuscript have been incorporated by using yellow background. We believe our manuscript is now stronger as a result of these modifications. An itemised point-by-point response to the Reviewers’ comments is presented below.
